# Hereditary Transthyretin Amyloidosis (hATTR) with Polyneuropathy Clusters Are Located in Ancient Mining Districts: A Possible Geochemical Origin of the Disease

**DOI:** 10.3390/biom14060652

**Published:** 2024-06-03

**Authors:** Per M. Roos, Sebastian K. T. S. Wärmländer

**Affiliations:** 1Institute of Environmental Medicine, Karolinska Institutet, 17177 Stockholm, Sweden; 2Department of Physiology, St. Göran Hospital University Unit, 11281 Stockholm, Sweden; 3Chemistry Section, Stockholm University, 10691 Stockholm, Sweden; 4CellPept Sweden AB, Kvarngatan 10B, 11847 Stockholm, Sweden

**Keywords:** neurodegeneration, protein aggregation, proteinopathy, metal–protein binding, metal toxicity, Familial Amyloid Polyneuropathy (FAP), amyloid

## Abstract

Hereditary transthyretin amyloidosis (hATTR) with polyneuropathy (formerly known as Familial Amyloid Polyneuropathy (FAP)) is an endemic amyloidosis involving the harmful aggregation of proteins, most commonly transthyretin (TTR) but sometimes also apolipoprotein A-1 or gelsolin. hATTR appears to be transmitted as an autosomal dominant trait. Over 100 point mutations have been identified, with the Val30Met substitution being the most common. Yet, the mechanism of pathogenesis and the overall origin of hATTR remain unclear. Here, we argue that hATTR could be related to harmful metal exposure. hATTR incidence is unevenly distributed globally, and the three largest defined clusters exist in Japan, Portugal, and Sweden. All three disease regions are also ancient mining districts with associated metal contamination of the local environment. There are two main mechanisms for how harmful metals, after uptake into tissues and body fluids, could induce hATTR. First, the metals could directly influence the expression, function, and/or aggregation of the proteins involved in hATTR pathology. Such metal–protein interactions might constitute molecular targets for anti-hATTR drug design. Second, metal exposure could induce hATTR -associated genetic mutations, which may have happened several generations ago. These two mechanisms can occur in parallel. In conclusion, the possibility that hATTR could be related to metal exposure in geochemically defined regions deserves further attention.

## 1. Introduction

Hereditary transthyretin amyloidosis (hATTR) with polyneuropathy (formerly known as Familial Amyloid Polyneuropathy (FAP)) is an endemic amyloidosis with variable clinical presentations characterised by the aggregation of proteins into harmful amyloid material. The most common aggregating protein is transthyretin (TTR), but aggregates of apolipoprotein A-1 and gelsolin are also observed [1]. For hATTR, the initial symptoms are loss of pain and temperature sensation in the feet caused by a length-dependent neuropathy, evolving into muscle weakness and paresis of feet and legs, autonomic failure with gastrointestinal symptoms, cardiomyopathy [1], sexual dysfunction and finally anal sphincter weakness [2]. The first symptoms appear at about 30 years of age, and the survival time is highly variable but may be, on average, ten years from diagnosis [1]. As TTR is mainly produced in the liver, orthotopic liver transplantations have been performed in an attempt to eliminate the source of TTR, with variable success [3]. The US Food and Drug Administration has approved several drugs against hATTR [4] such as the following: tafamidis, which blocks the rate-limiting step of TTR aggregation by inhibiting TTR tetramer dissociation [5]; patisiran, which inhibits the expression of both the variant and wt-TTR [6]; and inotersen, antisense oligonucleotides which cause degradation of the mutant and wt-TTR mRNA [7].

Hereditary transthyretin amyloidosis with polyneuropathy appears to be transmitted as an autosomal dominant trait. The most common mutation is the Val30Met substitution, although more than 100 other amyloidogenic point mutations [8] have been described at various geographic locations across the world (Figure 1). Yet some individuals with the TTR mutation never develop the clinical characteristics associated with hATTR [9], and a few patients with hATTR do not show any family history of the disease [9]. To what extent an environmental component contributes to hATTR seems unresolved [10]. One study has reported that genetic variants located in the non-coding regions of the *TTR* gene affect the clinical outcome of the disease [11]. Even though significant progress has been made in understanding—and treating—hATTR at the molecular level, the mechanism of pathogenesis and the overall origin of the disease remain unclear [12,13].

It has previously been argued that some neurodegenerative diseases could be related to metal exposure [14,15,16,17], possibly of geochemical origin [18,19]. Here, we investigate and discuss if such metal exposure could be a potential contributing factor for hATTR onset.

## 2. Materials and Methods

In this study, we investigate the hypothesis that hATTR incidence could be related to geochemical properties of different geographic locations. The approach is straight-forward. First, we review previous studies to identify the largest geographical clusters of hATTR incidence, which are unevenly distributed over the planet. Then, we review earlier studies to characterise the geochemical properties at these locations. Finally, we compare the geochemistry at the different hATTR clusters and attempt to find common features that could be related to hATTR pathogenesis.

## 3. Results

The incidence of hATTR is known to be unevenly distributed over various geographic regions (Figure 1). There is a general agreement in the published literature that the three largest defined hATTR clusters are located in Japan, Northern Portugal, and Northern Sweden [1,8,20,21]. Interestingly, all these three regions are ancient mining districts. In our opinion, this is probably not a coincidence. Below, we present details of these hATTR clusters and the geochemistry at their locations.

### 3.1. Japan

In Japan, three distinctive endemic hATTR foci have been described as follows: Arao city in the Kumamoto prefecture, Ogawa village in the Nagano prefecture, and Noto peninsula in the Ishikawa prefecture [8]. Japan is perforated with coal mines and mines for metals such as As, Au, Cd, Cr, Cu, Fe, Hg, Mn, Ni, Pb, Sn, U, and W. Specifically, Arao has been a mining town since the 18^th^ century with seven mines within the city itself, and Arao also has the largest coal mine harbour in Japan, i.e., the Port of Miike. The small mountain village of Ogawa is located at the confluence of several water streams draining the surrounding volcanic slopes. The Noto peninsula hosts the historical Togi metal mining fields [22] together with the Searashi Mn mine in the volcanic regions of the Noto peninsula, Ishikawa prefecture. The geological rationale for all these mines is that Japan is located where the Eurasian tectonic plate met the Izanagi plate some 70 million years ago. It should be noted that coal mining is associated with the release of metals with genotoxic properties [23], often found in elevated concentrations in coal mine tailings and waste, and leaching into agricultural soils typically in a concentration sequence of As > Zn > Ni > Cd > Cr > Cu > Pb [24]. Coal often also contains U [25] with known neurotoxic and genotoxic properties [26].

### 3.2. Portugal

In Portugal, hATTR is known as *Mal dos pèsinhos* or Foot-disease, which originally was described among fishermen from Póvoa de Varzim, north of Porto [21]. However, the entire northern coast of Portugal has presented cases, and this is the location of the largest hATTR cluster in the world [1]. Portugal is located in a geological zone with diverse mineral deposits and possible tectonic activity [27], where deposits of Sn, U, and W—tungsten (W) often follows tin ores—are present in the northern regions. Mining activities in Northern Portugal have been documented since prehistoric time [28,29,30]. The extensive Iberian tin belt concentrates in Northwestern Portugal, where the rivers flowing through the Iberian tin-rich areas follow a path westward with their lower course in Portuguese territory, eventually reaching the Atlantic coast [28,29]. Abandoned U mines are located east of Póvoa de Varzim, where runoffs may contaminate the groundwater and drain into the ocean. The environmental exposure is likely increased by the use of agricultural fertilisers, which are known to increase the solubility of U and its leach into the environment [31]. Thus, elevated concentrations of radionuclides in fish muscle have been detected, relating to past U mining in the rivers of Portugal [32].

The Atlantic Ocean west of Portugal is a seismically active zone where the Azores-Gibraltar Transform Fault Zone (AGFZ) has produced several large-magnitude earthquakes and volcanoes. In the original description of hATTR from Portugal [21], the index case fisherman describes the ingestion of large quantities of sea water in a drowning accident before deterioration in hATTR. No data on metal concentrations in the ocean above the AGFZ exist, but measurements in the Ave river flowing through Povoa de Varzim into the ocean have shown significantly elevated concentrations of neurotoxic metals [33]. To what extent these high metal concentrations emanate from volcanic rock along the AGFZ, from mining activities in Northern Portugal, or from metal exposure from textile, tanning, plastic, metal plating and rubber industries in the region remains to be elucidated. Either way, the possibility that Portuguese hATTR fishermen families have been subject to metal exposure from the waters west of Povoa de Varzim should be further explored.

The hATTR foci in Brazil and Mallorca—4th and 5th largest in the world—might be related to the large hATTR cluster in Portugal. Indeed, Portuguese ancestry has been found in almost all the Brazilian cases, which display overall the same disease characteristics as the most common Portuguese FAP type I patients, including the Val30Met mutation [34,35]. This mutation also dominates in the patients from Mallorca, which could have Portuguese origins [36,37]. However, Mallorca and Menorca were once mining islands [38,39,40]. Thus, independent origins of the Val to Met mutation, possibly induced by metal exposure, should not be excluded [41].

### 3.3. Sweden

In Sweden, a distinctive hATTR cluster exists in the northern municipality of Skellefteå [1,20,42], where mining activities for Ag, As, Au, Cu, Pb, and Zn have been ongoing for several centuries. Twenty-eight different metal mines have been opened in the Skellefteå mining district, which hold massive sulphide deposits hosted in metavolcanic rocks. Uranium minerals are also present [43]. Sweden is located in the region of Baltic bedrock and is therefore geologically stable. But in a palaeotectonic context, the Skellefteå district has been a collision site for tectonic plates [43], akin to the situation in Japan, which explains the large amounts of minerals present. In addition, local clustering of hATTR families has been described in the municipalities of Piteå and Lycksele, close to Skellefteå [20,44,45]. These cases might relate to common ancestors in Skellefteå but could also be connected to historic mining activities in the Piteå (Fe, Ag) and Lycksele (Zn, Cu, Ag, Au) regions.

## 4. Discussion and Conclusions

The fact that the three largest hATTR clusters are located in ancient mining districts suggest that hATTR could be related to harmful metal exposure. Specifically, the described clusters are located in seaside mining districts, where major metal exposure routes are enteric exposure by ingestion of fish and by drinking of well water [46]. There are two main mechanisms for the possible biological effects of metal exposure in hATTR, after its uptake into tissues and body fluids. In the first scenario, the metals directly affect the expression, function, and/or aggregation of the proteins involved in hATTR pathology, i.e., TTR, apolipoprotein A-1, and gelsolin. In the second scenario, metal exposure induces the mutations associated with hATTR, such as the TTR Val30Met mutation. A combination of these two mechanisms is possible. Both mechanisms could be tested at the molecular level with future biochemical experiments. In the second scenario, the disease-related mutations may have occurred several generations ago. The general conclusion would then be that exposure to neurotoxic and genotoxic metals from mining activities can induce adverse health effects. This would not be a novel observation. In the first scenario, the damaging metal–protein interactions are continually ongoing in hATTR patients. The primary line of defence against such possibly damaging interactions is the prevention of metal exposure. However, in already exposed individuals, these metal–protein interactions may constitute molecular targets for anti-hATTR drug design. It should be noted that this suggested explanatory model, which involves exposure to metals with neurotoxic and genotoxic properties, also takes into account the previously observed and yet unexplained huge variation in penetrance between foci [44,45], as well as the wide variation in age of onset for identical twins [42,47] and neighbouring foci [20]. In conclusion, the possibility that hATTR might be related to metal exposure in geochemically defined regions deserves further attention.

## Figures and Tables

**Figure 1 biomolecules-14-00652-f001:**
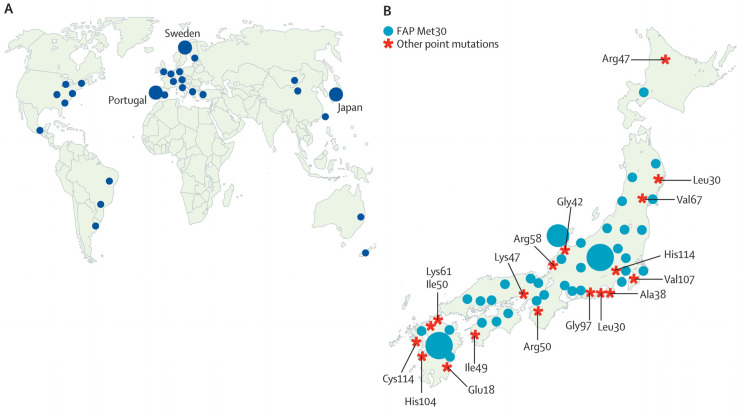
(**A**) Distribution of hATTR clusters across the world. (**B**) Distribution of hATTR cases in Japan, together with observed point mutations in the TTR protein. Image by PR, based on the information in Figures 1 and 2 in Araki and Ando, 2010 [8].

## Data Availability

All datasets generated and/or analysed during the current study are included in the manuscript.

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
