# Peer review of "Hereditary Transthyretin Amyloidosis (hATTR) with Polyneuropathy Clusters Are Located in Ancient Mining Districts: A Possible Geochemical Origin of the Disease"

_biomolecules, 2024, doi:10.3390/biom14060652_

Round 1

Reviewer 1 Report

Comments and Suggestions for Authors

The authors present an interesting theory of a geochemical origin of hereditary transthyretin amyloidosis in the larger clustering areas of the world. There might be an association here, but there are some things that need to be revised before the manuscript is ready for publication. Please see my more specific comments below:

1. FAP is an old term for the disease, please follow the latest nomenclature recommendations from the International Society of Amyloidosis throughout the paper.

2. TTR FAP (or ATTRv amyloidosis) is probably one of the most common amyloidoses. However, apo-A1 and gelsoline amyloidosis are separate and more rare diseases caused by other precursor proteins. Please update the Introduction and Discussion sections accordingly.

3. Introduction: sexual dysfunction is usually an early symptom of the disease, whereas anal sphincter dysfunction is a late symptom.

4. Introduction: to date, several other drugs than tafamidis are approved for treatment of ATTR amylodiosis.

5. Materials and Methods: It is true that Portugal, Sweden and Japan are considered as the main clustering areas of ATTRV30M amyloidosis in the world, but there are also quite large clusters in Mallorca and Brazil. Please consider including also these areas in the analysis.

6. Results: The Skellefteå area is the "hot-spot" for ATTRV30M amyloidosis in Sweden, but there are also a number of families in Piteå and Lycksele. It seems that the disease onset is generally earlier in the Skellefteå area than in the others. The Piteå and Lycksele families are probably descendants from the common ancestor i Skellefteå but, if applicable, you may also want to include some details on the mining activities in Piteå and Lycksele here.

7. Discussion and conclusion: The most recent common ancestor (MRCA) has been traced back to the 17th century for the Swedish ATTRV30M trait and to the 15th century for the Portuguese trait. As far as we know the Swedish trait is separate from the Portuguese, whereas Portugal has a common MRCA with Brazil and Japan. Thus, the disease probably occured before the mining era in these areas. Please consider updating the section based on this information.

8. Discussion and conclusion: ATTRV30M amyloidosis is inherited in an autosomal dominant pattern and there is a 50% risk for each child to inherit the TTR mutation from the affected parent. However, the penetrance of the disease varies between regions but also within families for yet unknown reasons. Both genetic and enviromental factors probably play a role for this variation in age of onset and  disease progression. Mining and metal contamination could play a role here, but more likely for the disese penetrance and not for the development of de novo TTR mutations based on the current evidence.

Author Response

  1. FAP is an old term for the disease, please follow the latest nomenclature recommendations from the International Society of Amyloidosis throughout the paper.
  • Point taken. Thank you. Nomenclature now adheres to the recommendations by the International Society of Amyloidosis (ISA) Nomenclature Committee. The term Hereditary transthyretin amyloidosis (hATTR) with polyneuropathy is now used throughout the manuscript.
  1. TTR FAP (or ATTRv amyloidosis) is probably one of the most common amyloidoses. However, apo-A1 and gelsoline amyloidosis are separate and more rare diseases caused by other precursor proteins. Please update the Introduction and Discussion sections accordingly.
  • We agree. The updated manuscript states both in the introduction and the discussion that FAP/hATTR might be caused by either of TTR, apo-A1 or gelsolin.
  1. Introduction: sexual dysfunction is usually an early symptom of the disease, whereas anal sphincter dysfunction is a late symptom.
  • Thank you. Modified accordingly. New wording: “…sexual dysfunction and finally anal sphincter weakness”.
  1. Introduction: to date, several other drugs than tafamidis are approved for treatment of ATTR amylodiosis.
  • All three FDA so far approved drugs are now covered in the Introduction. New sentence: ”The US Food and Drug Administration has approved several drugs against hATTR [4], such as Tafamidis, which blocks the rate‐limiting step of TTR aggregation by inhibiting TTR tetramer dissociation [5], Patisiran, which inhibits the expression of both variant and wt‐TTR [6], and Inotersen, antisense oligonucleotides which cause degradation of mutant and wt‐TTR mRNA [7].” Two new references have been added, one for patisiran and one for inotersen.
  1. Materials and Methods: It is true that Portugal, Sweden and Japan are considered as the main clustering areas of ATTRV30M amyloidosis in the world, but there are also quite large clusters in Mallorca and Brazil. Please consider including also these areas in the analysis.
  • Thank you for the suggestion. The clusters in Mallorca and Brazil have now been included in the manuscript.
  1. Results: The Skellefteå area is the "hot-spot" for ATTRV30M amyloidosis in Sweden, but there are also a number of families in Piteå and Lycksele. It seems that the disease onset is generally earlier in the Skellefteå area than in the others. The Piteå and Lycksele families are probably descendants from the common ancestor i Skellefteå but, if applicable, you may also want to include some details on the mining activities in Piteå and Lycksele here.
  • We agree. An extension into the municipalities of Piteå and Lycksele has been added, with new references: “In addition, local clustering of hATTR families has been described in the municipalities of Piteå and Lycksele, close to Skellefteå [19, 35, 36]. These cases might relate to common ancestors in Skellefteå, but could also be connected to historic mining activities in the Piteå (Fe, Ag) and Lycksele (Zn, Cu, Ag, Au) regions.”
  1. Discussion and conclusion: The most recent common ancestor (MRCA) has been traced back to the 17th century for the Swedish ATTRV30M trait and to the 15th century for the Portuguese trait. As far as we know the Swedish trait is separate from the Portuguese, whereas Portugal has a common MRCA with Brazil and Japan. Thus, the disease probably occurred before the mining era in these areas. Please consider updating the section based on this information.
  • All three areas are rich in valuable and easily accessible minerals, and have likely been mined for many centuries or even millennia. For Portugal, it is well known that the tin mines in the region of interest were the main reason for this region being conquered by the Roman empire. References for this are now provided in the updated manuscript. For the foci in Sweden and Japan, it appears that less archaeological information about prehistoric activities is available. This is something that could be further explored in future research.
  1. Discussion and conclusion: ATTRV30M amyloidosis is inherited in an autosomal dominant pattern and there is a 50% risk for each child to inherit the TTR mutation from the affected parent. However, the penetrance of the disease varies between regions but also within families for yet unknown reasons. Both genetic and enviromental factors probably play a role for this variation in age of onset and  disease progression. Mining and metal contamination could play a role here, but more likely for the disese penetrance and not for the development of de novo TTR mutations based on the current evidence.
  • Agree. Mining and metal could play a role in relation to penetrance. New sentence covering this aspect inserted: “It should be noted that this suggested explanatory model, which involves exposure to metals with neurotoxic and genotoxic properties, also takes into account the previously observed and yet unexplained huge variation in penetrance between foci [35, 36], as well as the wide variation in age of onset for identical twins [33, 38] and neighbouring foci [19].”

Reviewer 2 Report

Comments and Suggestions for Authors

This manuscript describes the hypothesis of the authors on the relationship between FAP and metal exposure. The hypothesis is interesting and further examination is expected. However, unfortunately, the manuscript is too descriptive and there is no evidence at the biomolecule level. Therefore, this reviewer recommend the authors to submit this manuscript to some appropriate journal.

Author Response

The reviewer is correct that there is currently no molecular-level evidence. But collecting such evidence is a massive undertaking, and clearly a separate project. The relevance of the current manuscript for this journal is that the manuscript describes a clear hypothesis that can be experimentally tested at the molecular level. This is now clearly stated in the discussion section of the manuscript.

Reviewer 3 Report

Comments and Suggestions for Authors

The process of amyloid formation in general, and TTR amyloid in particular, is very complex and involves many factors, of which only one was chosen. If this one factor is the main cause of all others, then such correlations would have to be proven. No consideration was given to differences in the mitochondrial DNA haplotype of residents of regions where metal concentrations are postulated to affect disease prevalence compared to populations where FAP prevalence is at lower levels. There is a lack of analysis and data on the effects of specific metals and their chemical forms on TTR amyloid formation. The article fails to consider and address many important aspects of TTR aggregation. Mutation in the TTR molecule is only one of the causes of amyloid deposits, aging and proteolytic digestion processes have been completely omitted. As for statistics, they too are very rough and approximate.  For example, there is no accurate, detailed comparative data on the occurrence and content of ions/ different chemical compounds of these metals in different places on earth, distinguishing between their content in water and soil. One scheme to draw the conclusion is not enough. It is a good starting point but there should be more detailed data that needs to be carefully analysed to be able to draw such  conclusions as the authors did.

Author Response

We do not understand this review comment. In our short communication we argue that harmful metal exposure is a possible factor in FAP/hATTR etiology that should be further explored, and that toxic metal exposure might be able to induce the Val→Met mutation observed in many FAP/hATTR patients. What kind of detailed data does the reviewer think would be required to make those arguments?

Reviewer 4 Report

Comments and Suggestions for Authors

The authors make an interesting observation that FAP cases cluster around ancient mining districts and propose that toxic metal exposure from the environment is a contributing factor towards development of FAP. Although the manuscript puts forward a causality hypothesis, the evidence cited is purely correlational and demands more rigorous analysis to prove causality. In addition, I have the following comments:

It is proposed by the authors that certain mutations of the transthyretin gene (e.g. V30M) can be caused by metal exposure several generations ago. I am curious how metal exposure could lead to a specific mutation or a set of mutations specific for a particular disease, that too in the germline. This sounds too far-fetched and scientifically implausible.

Out of all the mining districts in the entire world, what makes these mining districts unique that clusters of FAP are found surrounding these districts?

The authors could not point to any specific metal as the major culprit. However, they suggest that leaching of such metals in the soil as a causative factor for chronic exposure. Is this statement supported by evidence such as increased prevalence of arsenicosis or congenital malformations in such regions?

“…in already exposed people these particular metal-protein interactions would constitute molecular targets for anti-FAP drug design.” – What is the evidence behind metal-protein interactions playing a role in TTR aggregation?

Overall, the paper consists of a curious observation only but the rigorous data analysis to prove that such an association does not occur purely by chance and has a causal relationship is completely missing. The authors may find inspiration in their cited paper DOI: 10.1007/s00415-023-11888-8 for such a critical analysis.

Author Response

The reviewer is correct that the manuscript presents correlations, and that more rigorous analysis is required to prove causality. However, the aim of the paper is not to causally prove the origin of the disease. This is a short communication where the aim is to argue an interesting aspect of the disease, in the form of important but so far neglected correlations that deserve to be further explored. We are very thankful for the suggestions about metal-protein interactions, metal-induced mutations, correlations with arsenicosis or congenital malformations, et cetera. All of these are valid questions that can´t be pursued in this paper, but they will be important in the future research that will investigate the different aspects of the basic hypothesis that is presented in the current short communication.

Reviewer 5 Report

Comments and Suggestions for Authors

In this study, the authors investigated and discussed if such metal exposure could be a possible contributing factor for FAP onset. Accordingly, FAP could be related to metal exposure in geochemically defined regions deserves further attention. The introduction provided sufficient background and include all relevant references. The conclusions supported by the results.

Author Response

Thank you. We consider this article a possible starting point for further discussions on the role of geochemical metal exposure in relation to FAP/hATTR pathogenesis, of interest to the informed readers of Biomolecules.

Reviewer 6 Report

Comments and Suggestions for Authors The authors submitted a short communication where they discussed on the relationship between FAP and harmful metal exposure, and hypothesized two potential hypotheses or mechanisms for how harmful metals could induce FAP. The topic is certainly interesting: clarifying the mechanisms of pathogenesis and overall origin of FAP is certainly important considering the enormous impact that this pathology has on individuals and on society. The work is well written. The methods are sound event though, given the nature of the short communications, are not presented in details. There are not English errors/typos. Always due to the nature of the short communications, the authors did not present tools to demonstrate what they want to say.

Author Response

Thank you. We think this article should stay within the formal limitations of a Short Communication, thereby hopefully sparking further discussions on the role of genotoxic and neurotoxic metals in relation to FAP/hATTR

Round 2

Reviewer 1 Report

Comments and Suggestions for Authors

The authors have revised the manuscript according to the reviewers' comments.

Reviewer 2 Report

Comments and Suggestions for Authors

I previously made a decision based on the fact that this manuscript did not have any evidences of molecular basis. However, the manuscript has been modified extensively and, as authors said, further studies will be performed at the molecular level guided by the discussion of this manuscript.

Although I worried that this manuscript may be out of the journal scope of Biomolecules, this manuscript has been improved and possible be published in this journal, if editors think that there is no problem between the journal scope and this revised manuscript.

Reviewer 4 Report

Comments and Suggestions for Authors

Thank you for the significantly improved updated manuscript. I understand the scopes of the communication type paper and the current version is enough for that. I also appreciate the addition of the clusters of cases in Brazil and Mallorca and the correlation with Portuguese ancestry.

While building on the hypothesis that the clinical heterogeneity of TTR V30M mutation could be due to environmental exposure (Lines 47-52), the authors may want to include an important finding where it was shown that noncoding variants of TTR gene correlate with clinical outcome (https://doi.org/10.1038/ejhg.2017.95).

In addition, I recommend a few typographical corrections as follows:

1.      Line 43-46: First letters of generic drugs should not be capitalized.

2.      Line 47: The name of the disease should be written as “hATTR” and not “hATTR disease”

Reviewer 6 Report

Comments and Suggestions for Authors

The authors believe that this article should remain within the formal limits of a brief communication, hopefully sparking further discussion on the role of genotoxic and neurotoxic metals in relation to FAP/hATTR.

If the Editor also agrees with this decision, I have no further comments on the matter, so the article may be suitable for publication.

I have no further comments on this matter, so the article may be suitable for publication.